# Efficacy and safety of tibial cortex transverse transport for diabetic foot: A protocol for systematic review and meta-analysis

**Jiaxing Guo**[1☯], **Huhe Bao**[2☯], **Lideer**[3], **Xiyu Ni**[3], **Yaxin Zhao**[3,4], **Guanwen Sun** [2]*

**1** Department of Arthrosis, The Second Affiliated Hospital of Inner Mongolia Medical University, Huhhot, Inner Mongolia Autonomous Region, China, **2** Department of Trauma and Orthopedics, Inner Mongolia People's Hospital, Hohhot, Inner Mongolia Autonomous Region, China, **3** Graduate School, Inner Mongolia Medical University, Hohhot, Inner Mongolia Autonomous Region, China, **4** Department of Endocrinology, Inner Mongolia People's Hospital, Hohhot, Inner Mongolia Autonomous Region, China

☯ These authors contributed equally to this work.
* sunguanwen@sina.com

## Abstract

### Introduction

Diabetic foot (DF) is one of the most serious chronic complications of diabetes. In recent years, the use of the tibial cortex transverse transport (TTT) technique has enabled great progress in microcirculation reconstruction and achievement of good outcomes in DF treatment. The objective of this systematic review protocol is to evaluate the efficacy and safety of TTT for DF.

### Methods

Literature search was conducted using the Cochrane Library, Embase, PubMed, Web of Science, China Science Technology Journal Database (VIP), Wanfang Data, China National Knowledge Infrastructure (CNKI), Chinese Biomedical Literature Service System (SinoMed), and Chinese Biomedical Literature Service System (CBM) from inception until March, 1st 2022. In addition, our reviewers will retrieve dissertations, grey literature, systematic reviews, and reference lists of the relevant studies. Randomized controlled trials (RCTs) which compared the TTT for DF with conventional treatment will be included. Our reviewers will perform subgroup analysis, sensitivity analysis, and publication bias analysis to evaluate the heterogeneity and robustness. RevMan 5.3 software and Stata V.16.0 software will be used to analyze the available data.

### Ethics and dissemination

Ethical approval was not required because this protocol neither collected private information, nor involved animal experiments. The research was disseminated by academic journals or related meetings.

**Data Availability Statement:** Since this paper is a protocol for systematic review and meta-analysis and the relevant content retrieval and data

processing have not been completed, we did not upload the data to a public repository. All previous search criteria and contents can be found in the manuscript and supplementary files. When we complete all data collection and processing, we will upload relevant data to a public repository and submit the final meta-analysis to the PLOS ONE.

**Funding:** GWS, The Doctoral Research Start-up Fund of Inner Mongolia People's Hospital(grant number 2020BS01) http://www.nmgyy.cn/. The funder had and will not have a role in study design, datacollection and analysis, decisiontopublish, orpreparation of the manuscript.

**Competing interests:** The authors have declared that no competing interests exist.

## PROSPERO registration number

CRD42021279717.

## Introduction

Diabetic foot (DF) manifests as skin infection, ulcers, and deep tissue destruction around the ankle. It represents one of the most serious chronic complications of diabetes [1, 2]. Pathological changes in DF mainly involve lower extremity vasculopathy and neuropathy [3]. The global prevalence of DF is 6.3%, and about 25% of diabetic patients will develop foot ulcers [4]. DF is a complex surgical problem because of its high rates of recurrence, morbidity, disability, and mortality. At present, the conventional surgical treatments for DF include vacuum-sealing drainage, local debridement, free skin flap grafting, vascular bypass, and interventional therapy [5–7]. However, in patients with extensive and severe vascular lesions, conventional approaches often did not achieve the anticipated results, and patients were exposed to a high risk of amputation [8]. In recent years, the use of the tibial cortex transverse transport (TTT) technique has enabled great progress in microcirculation reconstruction and achievement of good outcomes in the therapeutic management of DF. TTT technology can improve the microcirculation of the affected limb, promote the recovery of nerves and repair refractory ulcers, thus finally achieving the goal of treating DF [9, 10].

TTT is based on Ilizarov's "tension-stress rule" and "natural rebuilding regeneration theory". Tibial draft produces tension stress, which stimulates cell proliferation and the synchronous growth and repair of bones and associated muscle, fascia, blood vessels, and nerves, improves tissue microcirculation, and promotes the healing of leg and foot ulcers [11, 12]. Ilizarov conducted the traction osteogenesis test on a canine leg and found that slow and continuous traction of the affected limb in the surgical area can greatly improve the regeneration and reconstruction of blood vessels as well as microcirculation [13]. Barker et al. demonstrated that traction promoted the growth of bones, nerves, blood vessels, and muscles in patients with nonunion fractures [14]. Long Qu was the first to use TTT to treat lower extremity thromboangiitis obliterans [15]. Later, Qikai Hua improved the preexisting surgical protocol and was the first to apply it in the treatment of DF [16]. The healing time of TTT for severe DF was 3–12 months, the average healing time was 5–6 months, and the recurrence rate was 4.5%–7.7% within 1 year [17]. Qiqiu et al. performed the TTT treatment in 150 patients with DF, and found that the wounds of 144 patients fully healed, two patients with incomplete healing developed significant wound reduction, and four patients had to undergo toe amputations [18]. Traditional osteotomy has the disadvantages of long incision, large fenestration area and multiple complications [19]. With the rapid development of minimally invasive and rapid rehabilitation, the healing time of ulcer surface has been greatly shortened due to small incisions and appropriate reduction of transport time [20]. If data are sufficient, we will perform a subgroup analysis of modified TTT versus conventional TTT osteotomy. However, TTT technology is still in the development stage. In addition, postoperative bacterial contamination can easily lead to infection, skin flap necrosis, fracture nonunion, delayed wound healing, and serious fractures. Therefore, it is necessary to conduct a novel and comprehensive systematic review to evaluate the efficacy and safety of TTT for DF.

The underlying mechanism of TTT treatment for DF may be multifactorial. TTT was found to aid in reducing local inflammatory response and facilitating local stem cell activation and tissue repair [21]. M1 macrophages promote inflammatory responses, while M2

macrophages play a role in opposing inflammation and promoting tissue regeneration [22, 23]. Wei et al. compared tissue sections of patients after TTT and found that the M1/M2 ratio significantly decreased one month after surgery when compared to that before surgery. They assumed that TTT may act by transforming the macrophages into the M2-type, rebuilding the polarization balance, and promoting wound healing [24]. Shuanji et al. showed that TTT could significantly increase the expression of the vascular endothelial growth factor (VEGF), basic fibroblast growth factor (bFGF), epidermal growth factor (EGF), and the platelet derived growth factor (PDGF) in ulcer tissue, which may be the repair mechanism underlying DF wound healing [25].

At present, few systematic reviews have reported on the effectiveness of TTT for DF [26–28], however, the evidence levels in the conducted research are not high. This is due to the lack of high-quality randomized controlled trials (RCTs), inadequate outcome indicators, limited search scope, and other limitations. Therefore, we propose this systematic review protocol to evaluate the effectiveness of TTT in patients with DF.

## Methods

### Study registration

This protocol will be reported in accordance with the Preferred Reporting Items for Systematic Review and Meta-Analyses Protocols (PRISMA-P) guidelines [29]. This protocol was registered on PROSPERO under the CRD number CRD42021279717.

### Eligibility criteria

**Types of studies.**   In this systematic review, we only included the RCTs which evaluated the effectiveness and safety of TTT for DF. Basic or animal studies, literatures with incomplete data, and literature reviews were excluded. Case reports and conference articles were also excluded due to their low level of scientific evidence.

**Types of participants.**   Patients diagnosed with DF using any recognized diagnostic criteria will be included regardless of their gender, age, race, and the duration and severity of the disease.

**Types of interventions.**   Patients with DF who were treated with TTT, either alone or in any combination with conventional treatments, will be included. There were no restrictions on the time or method of bone transport. Both groups received the same basic treatment.

**Types of comparisons.**   Patients in the comparison group had undergone conventional treatment approaches which included artery bypass grafting, debridement, vacuum-sealing drainage, flap reconstruction, vascular intervention, drug therapy, and related adjuvant therapy.

**Types of outcomes.**   The main outcomes will be healing time and effective rate. The secondary outcomes will include visual analogue scale, ankle brachial index, bFGF, VEGF, amputation rate, ankle skin temperature, hospitalization time, healing rate, blood flow rate of dorsalis pedis and complications, such as vascular thrombosis, healing delay, pin tract infection, incision infection, tibial shaft fracture, and osteomyelitis.

### Information sources and search strategy

Only English and Chinese articles were included in this study. The English databases included Cochrane Library, Embase, PubMed, Web of Science. The Chinese databases included China Science Technology Journal Database (VIP), Wanfang Data, China National Knowledge Infrastructure (CNKI), Chinese Biomedical Literature Service System (SinoMed), and Chinese

Biomedical Literature Service System (CBM). The search time ranged from inception to March 1st, 2022.

The search terms included "tibial transverse transport," "transverse tibial bone transport," "tibial cortex transverse transport," "TTT," "Ilizarov technique," "Ilizarov method," "Ilizarov external fixation," "Ilizarov," "diabetic foot," "diabetic feet," "diabetic angiopathies," "diabetic," "foot ulcer," "DF," "randomized controlled trial," "randomized," "randomly," and "RCT." Our reviewers will also search ongoing or unpublished trials from ClinicalTrials.gov, International Clinical Trials Registry Platform, and Chinese Clinical Trial Registry. To avoid missing any other relevant studies, we retrieved dissertations, gray literature, systematic reviews and reference lists of relevant studies. Table 1 shows an example of the PubMed search strategy.

## Study selection

Two reviewers will insert the retrieved results of the searched studies into the Endnote X9 software and exclude duplicated studies. Then, they will investigate the title and abstract to exclude the studies that do not meet the inclusion criteria. Finally, they will include the eligible studies after reading the full text of the remaining studies. If two reviewers have differing opinions, they will carry out a discussion with the third reviewer and make a final decision. The process of study selection has been shown using a flow chart (Fig 1).

## Data extraction

Two reviewers will use a standard data extraction table to extract the necessary data from the included articles. The following specific information from the included articles will be extracted: (a) study characteristics including title, first author name, publication year, countries, article type, inclusion and exclusion criteria, and duration of the follow-up, (b)

**Table 1. Search strategy for the PubMed.**

| Number | Search terms |
|---|---|
| #1 | tibial transverse transport.ti, ab. |
| #2 | transverse tibial bone transport.ti, ab. |
| #3 | tibial cortex transverse transport.ti, ab. |
| #4 | TTT.ti, ab. |
| #5 | Ilizarov technique. mesh. |
| #6 | Ilizarov method.ti, ab. |
| #7 | Ilizarov external fixation.ti, ab. |
| #8 | Ilizarov.ti, ab. |
| #9 | OR #1-#8 |
| #10 | diabetic foot. mesh. |
| #11 | diabetic feet.ti, ab. |
| #12 | diabetic angiopathies. mesh. |
| #13 | diabetic.ti, ab. |
| #14 | foot ulcer. mesh. |
| #15 | DF.ti, ab. |
| #16 | OR #9-#15 |
| #17 | randomized controlled trial. mesh. |
| #18 | randomized.ti, ab. |
| #19 | randomly.ti, ab. |
| #20 | RCT.ti, ab. |
| #21 | OR #17-#21 |
| #22 | #9 AND #16 AND #21 |

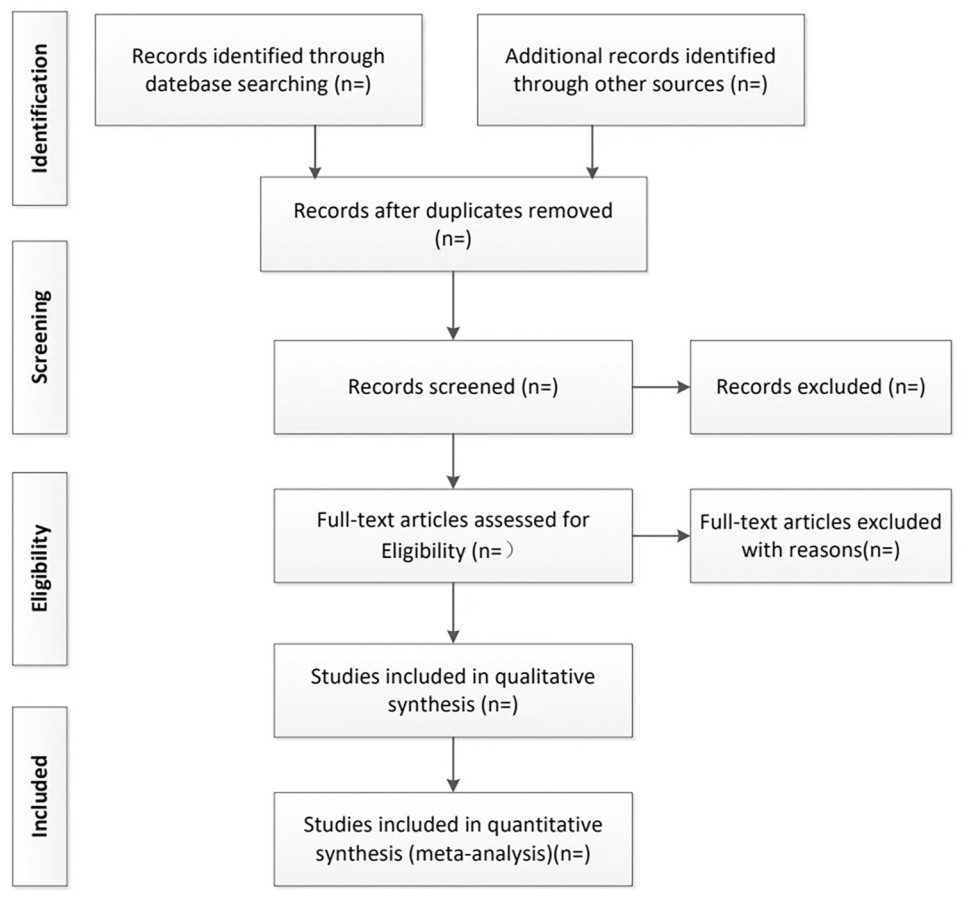

**Fig 1. Flowchart of study selection.**

participant characteristics including sex, race, age, diagnostic criteria, duration and severity of DF, Wagner grade, and sample size, (c) interventions and comparators including TTT type and program, external fixator type, and conventional treatment type, (d) main outcomes including the healing time and effective rate, (e) secondary outcomes including visual analogue scale, ankle brachial index, bFGF, VEGF, amputation rate, ankle skin temperature, hospitalization time, healing rate, blood flow rate of dorsalis pedis and complications (such as vascular thrombosis, healing delay, pin tract infection, incision infection, tibial shaft fracture, and osteomyelitis). If the data provided in the included study is unclear or missing, our reviewers will contact the corresponding author by email for clarification. If two reviewers have differing opinions, they will carry out a discussion with the third reviewer and make a final decision.

## Assessment of study quality

Two reviewers will use the Cochrane Handbook for Systematic Reviews of Interventions to assess the risk of bias in the included studies [30]. The evaluation standards mainly included the following seven aspects: random sequence generation, allocation hiding, blinding of participants and personnel, blinding of outcome assessment, incomplete outcome data, selective outcome reporting, and other biases. The rating of the item's quality will be based on three levels: high risk, unclear risk, or low risk. Any disagreement between two reviewers will be resolved through a discussion with the third reviewer.

## Statistical analysis

**Measures of treatment effect.** Two reviewers will independently use Stata V.16.0 software and Review Manager V.5.3 software to statistically analyze the efficacy data. For dichotomous outcomes, we will use either odds (OR) or risk ratios (RR) to measure the treatment effect. Inverse Variance will be used to study the dichotomous outcomes. The study confidence interval and the total confidence interval are both 95%. We will analyze the mean difference (MD) or standardized mean difference (SMD) with 95% CI for continuous outcomes. A $P < 0.05$ was considered statistically different.

**Assessment of heterogeneity.** $I^2$ statistic will be used to assess the degree of heterogeneity. A random effects model allows the true effect to vary across studies. It builds in an estimate of the between-study variation in effect size and typically provides wider overall confidence intervals [31]. The random effects model can be adopted irrespective of the outcome of the $I^2$ statistic for heterogeneity. It will be implemented using the standard DerSimonian-Laird (D-L) method. The confidence interval for $I^2$ is 95%.

**Subgroup analysis.** When the necessary data are available, we will investigate the source of heterogeneity using a subgroup analysis based on the Wagner degree, follow-up time, protocol of bone transport, type of conventional treatment and type of external fixator.

**Sensitivity analysis.** The sensitivity analysis for evaluating the reliability and robustness of our research will be carried out by including one study at a time. In order to avoid any valuable studies from being filtered out, we will include all studies that meet the inclusion and exclusion criteria. Whenever heterogeneity is detected, we do not ignore it, and we try to find its source. If conditions permit, a continuity correction factor will be added to each of the four cells of the 2×2 table for zero event studies, i.e., to the event and non-event in the treatment arms, and event and non-event in the control arms. The constant continuity factor will be 0.5.

## Assessment of publication bias

If more than 10 studies are included, we will use funnel plots and Egger's test to assess potential publication bias [32]. If publication bias does exist, we will further analyze it using the fill and trim method.

## Grading the quality of evidence

The Grading of Recommendations Assessment, Development and Evaluation (GRADE) will be used to assess the quality of evidence for the whole study [33].

## Discussion

DF is one of the main causes of disability and death in patients with diabetes. Additionally, its recurrence rate and medical costs are high. Worldwide, DF is ranked 10[th] of the diseases that cause heavy burdens to patients [34]. Conventional surgical treatments for DF include debridement, vacuum drainage, and vascular intervention. These methods for DF, such as interventional therapy, may cause serious adverse reactions. Therefore, an increasing number of surgeons worldwide have begun exploring TTT as a therapeutic approach for DF management. TTT is a unique surgical approach for DF and has attracted increasing attention due to its short operation time, few side effects, simple operation procedure, and high limb saving rate [35]. Several clinical trials have demonstrated its efficacy in DF, suggesting that TTT may play a significant role in the reconstruction of microcirculation.

The TTT technique is a complex intervention. Although its efficacy in the treatment of DF has been confirmed by some clinical trials, its exact effectiveness remains to be further

elucidated. The proposed systematic review aims to summarize the available evidence from RCTs and report the efficacy of TTT for DF. For the first time, we will evaluate the strength of evidence that was missing in the published systematic reviews based on the GRADE method. In conclusion, our study aims to assess the efficacy and safety of TTT for DF and provide reliable clinical evidence for the treatment of DF.

## Supporting information

**S1 Checklist. PRISMA-P 2015 checklist.**
(DOC)

## Author Contributions

**Conceptualization:** Jiaxing Guo, Guanwen Sun.

**Data curation:** Xiyu Ni.

**Formal analysis:** Huhe Bao, Yaxin Zhao.

**Funding acquisition:** Guanwen Sun.

**Methodology:** Huhe Bao,  Lideer, Xiyu Ni, Yaxin Zhao.

**Software:** Jiaxing Guo, Xiyu Ni.

**Supervision:**  Lideer, Guanwen Sun.

**Writing – original draft:** Jiaxing Guo, Huhe Bao,  Lideer.

**Writing – review & editing:** Jiaxing Guo, Huhe Bao, Guanwen Sun.

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
