## [Decision Letter · Decision Letter 0]

8 Jun 2022

PONE-D-21-40669Efficacy and safety of tibial transverse transport for diabetic foot: a protocol for systematic review and meta-analysisPLOS ONE

Dear Dr. Sun,

Thank you for submitting your manuscript to PLOS ONE. After careful consideration, we feel that it has merit but does not fully meet PLOS ONE’s publication criteria as it currently stands. Therefore, we invite you to submit a revised version of the manuscript that addresses the points raised during the review process.

The manuscript has been reviewed by several experts in the field.  While the findings are interesting, there remain deficiencies with the manuscript. The author is invited to revise and resubmit the manuscript.  The authors should respond to each of the comments. 

We look forward to receiving your revised manuscript.

Kind regards,

Dengshun Miao

Academic Editor

PLOS ONE

Journal Requirements:

4.  Thank you for stating the following in the Funding Section of your manuscript:

“This study was funded by the doctoral research start-up fund of Inner Mongolia People’s Hospital (2020BS01).”

We note that you have provided additional information within the Funding Section that is not currently declared in your Funding Statement. Please note that funding information should not appear in the Funding section or other areas of your manuscript. We will only publish funding information present in the Funding Statement section of the online submission form.

“The authors have declared that no competing interests exist.”

Reviewers' comments:

Reviewer's Responses to Questions

**Comments to the Author**

1. Does the manuscript provide a valid rationale for the proposed study, with clearly identified and justified research questions?

Reviewer #1: No

Reviewer #2: Partly

Reviewer #3: Yes

2. Is the protocol technically sound and planned in a manner that will lead to a meaningful outcome and allow testing the stated hypotheses?

Reviewer #1: No

Reviewer #2: No

Reviewer #3: Yes

3. Is the methodology feasible and described in sufficient detail to allow the work to be replicable?

Reviewer #1: Yes

Reviewer #2: No

Reviewer #3: Yes

4. Have the authors described where all data underlying the findings will be made available when the study is complete?

Reviewer #1: No

Reviewer #2: No

Reviewer #3: Yes

5. Is the manuscript presented in an intelligible fashion and written in standard English?

Reviewer #1: No

Reviewer #2: Yes

Reviewer #3: Yes

6. Review Comments to the Author

You may also provide optional suggestions and comments to authors that they might find helpful in planning their study.

Reviewer #1: In this paper, they described the potential protocol for the meta-analysis about the effects of tibial transverse transport for diabetic foot.

As the review of the specific topic, the paper seems to be superficial. Moreover, only the description of the protocol is not impressive. They did not show any results.

The clear definition of TTT is required. The abbreviation is not adequately explained, for example, VEGF, bFGF, EGF, PDGF. The description of reviewers, two reviewers, the third author seem to be odd as the subject in the manuscript writing. English editing by a native is necessary.

Reviewer #2: Dear Author,

The manuscript I received is incomplete. I found only abstract, introduction, statistical analysis, methods and discussion. I did not find the results or other data pertinent to the study. I suggest you submit the full manuscript for evaluation.

Reviewer #3: In this review, the authors raise a new, comprehensive analysis to validate the beneficial treatment of tibial transverse transport (TTT) in patients with diabetic foot. This evaluation is valuable, since the current literature do not reach to a level with clear outcome indicators and limited search scope. I believe this review will bridge this cap. One issue the author may need to address is how TTT treatment effectively functions in the different stage of diabetic foot.

7. PLOS authors have the option to publish the peer review history of their article (what does this mean?). If published, this will include your full peer review and any attached files.

Reviewer #1: No

Reviewer #2: No

Reviewer #3: No

---

## [Author Response · Author response to Decision Letter 0]

20 Jul 2022

Dear Reviewers:

Thank you for the valuable and helpful comments on our manuscript titled “Efficacy and safety of tibial transverse transport for diabetic foot: a protocol for systematic review and meta-analysis” (ID: PONE-D-21-40669). We have read your comments carefully and made corrections accordingly. We have uploaded the revised manuscript following the instructions provided in your letter. The revised sections are shown in red, and our responses to your comments are described below：

Reviewer #1:

1.Comment: As the review of the specific topic, the paper seems to be superficial. Moreover, only the description of the protocol is not impressive. They did not show any results.

Response: Thank you for pointing out this deficiency. We have revised and added content to the manuscript according to your advice. We defined diabetic foot (DF) and described pathological changes, epidemiological characteristics, and treatment status in the Introduction. Many traditional treatments are currently available for DF but have not achieved the expected results. As new technology, the tibial cortex transverse transport (TTT) technique has achieved satisfactory therapeutic results for DF. We defined TTT and discussed its advantages and relevant clinical studies that used this technology. In addition, the mechanism underlying TTT treatment for DF was systematically summarized. Many parts of the manuscript, such as eligibility criteria, information sources, search strategy, study selection, data extraction, assessment of study quality and publication bias, and statistical analysis, have been carefully revised and improved. We trust that these changes (shown in red) made the content more comprehensive and rigorous.

This paper is a protocol for systematic review and meta-analysis rather than original research. Data collection is ongoing, and relevant results will be published in the final meta-analysis. Literature on TTT treatment of diabetic foot is limited. This protocol can help fill knowledge gaps and establish guidelines for research design. Publishing research protocols allows researchers and funding agencies to keep abreast of research activities in related fields that are not yet widely known. In addition, these protocols improve research quality by reducing the number of duplicate studies and identifying deviations from the protocol during the study period. Please let us know if you have any questions or suggestions, and we will make corresponding modifications.

2.Comment: The clear definition of TTT is required.

Response: Thank you for your advice. TTT is a new technique for treating chronic lower-extremity angiopathy, such as that caused by diabetes. TTT has not been clearly defined to date. Notwithstanding, according to the most commonly accepted definition, TTT is based on Ilizarov’s "tension-stress rule" and natural rebuilding regeneration theory. Tibial draft produces tension stress, stimulating cell proliferation, stimulating the synchronous growth and repair of bones and associated muscle, fascia, blood vessels, and nerves, improving tissue microcirculation, and promoting the healing of leg and foot ulcers [11-12] (page 3-4, lines 64-68).

11. Ilizarov GA. The tension-stress effect on the genesis and growth of tissues: Part I. The influence of stability of fixation and soft tissuepreservation. Clin Orthop. 1989;238:249-281. https://doi.org/10.1097/00003086-198902000-00029 PMID: 2910611.

12. Chen Y, Kuang XC, Zhou J, Zhen PX, Zeng ZS, Lin ZX,et al. Proximal tibial cortex transverse distraction facilitating healing and limb salvage in severe and recalcitrant diabetic foot ulcers. Clin Orthop Relat Res. 2020;478(4):836-851. https://doi.org/10.1097/CORR.0000000000001075 PMID: 31794478

3.Comment: The abbreviation is not adequately explained, for example, VEGF, bFGF, EGF, PDGF. 

Response: We are grateful for the suggestion. These abbreviations have been properly defined. The revised sections are shown in red. (page 5, lines 106-109).

4.Comment: The description of reviewers, two reviewers, the third author seem to be odd as the subject in the manuscript writing.

Response: We apologize for the misunderstanding. Division of labor for authors is usually described in the meta-analysis to avoid conflicts related to work allocation and author cooperation and improve workflow and work efficiency. We described the reviewers' work assignment according to the general habit of meta-analysis. Thank you for your careful review. 

5.Comment: English editing by a native is necessary.

Response: Thank you for the suggestion. The manuscript was carefully revised by the authors for grammar and language issues and by Editage, a professional English language editing company. We trust that the methodology is better described in the revised manuscript. However, if you find other problems, please do not hesitate to let us know, and we will act accordingly to meet the standards of your journal.

Reviewer #2:

1.Comment: The manuscript I received is incomplete. I found only abstract, introduction, statistical analysis, methods and discussion. I did not find the results or other data pertinent to the study. I suggest you submit the full manuscript for evaluation.

Response: We apologize for the confusion. This paper is a protocol for systematic review and meta-analysis, rather than original research. Data collection is ongoing, and relevant results will be published in the final meta-analysis. This protocol can enable researchers and funding agencies to stay informed about research activities that are not widely publicized. These protocols improve research quality by reducing the number of duplicate studies. Moreover, full protocols provide more information than is required by trial registries, increasing research transparency and helping editors, reviewers, and readers identify deviations from the protocol during the study period.

Reviewer #3:

1.Comment: In this review, the authors raise a new, comprehensive analysis to validate the beneficial treatment of tibial transverse transport (TTT) in patients with diabetic foot. This evaluation is valuable, since the current literature do not reach to a level with clear outcome indicators and limited search scope. I believe this review will bridge this cap. One issue the author may need to address is how TTT treatment effectively functions in the different stage of diabetic foot.

Response: We appreciate your positive feedback. Our team is conducting clinical trials on TTT treatment of Wanger 3/4 diabetic foot to quantify the gene and protein expression of vascular endothelial growth factor (VEGF) and hypoxia-inducible factor 1-alpha (HIF-1α) during treatment and assess the effect of TTT on microvessel density (MVD) in diabetic foot wounds. Relevant manuscripts will be published after the completion of these experiments. We look forward to your suggestions.

We appreciate for reviewers’ warm work earnestly, and hope that the corrections will meet with approval. Some changes that we have made will not influence the content and framework of the paper, and we have thus not listed those changes, but marked them in red in the revised text. We trust that the revised manuscript is suitable for publication in your journal.

Sincerely,

Guanwen Sun

Email: sunguanwen@sina.com

---

## [Decision Letter · Decision Letter 1]

23 Aug 2022

PONE-D-21-40669R1Efficacy and safety of tibial cortex transverse transport for diabetic foot: a protocol for systematic review and meta-analysisPLOS ONE

Dear Dr. Sun,

Thank you for submitting your manuscript to PLOS ONE. After careful consideration, we feel that it has merit but does not fully meet PLOS ONE’s publication criteria as it currently stands. Therefore, we invite you to submit a revised version of the manuscript that addresses the points raised during the review process.

The manuscript has been seen by four reviewers. Unfortunately, upon inspection of the reviewer comments we noticed that the previous evaluation of Reviewer #4 has not been included in the first decision letter. I would to sincerely apologize for this mistake on our side. However, we consider their comments crucial to the manuscript  and would like to ask you to address their them before we can proceed. Please see the "Additional Editors comment" section for a copy of Reviewer #4's queries.

We look forward to receiving your revised manuscript and I would like to apologize again for not including all reviewer comments in the previous letter.

Kind regards,

Thomas Tischer

Staff Editor

PLOS ONE

Additional Editor Comments:

Comments from Reviewer #4:

I will focus on methods and reporting

Major

1) Define the outcomes that are not self explanatory. For example, I've never heard of "effective rate" before.

2) Avoid fixed effect models since they under-perform in the presence of ANY heterogeneity. Random-effects (RE) models are more conservative and provide better estimates with wider confidence intervals: http://www.ncbi.nlm.nih.gov/pubmed/11252006 and http://www.ncbi.nlm.nih.gov/pubmed/21148194 . What does the arbitrary 50% cut-off point add? Why is 49% fine and 51% an issue? My point is that a RE model will work better, in the presence of 5% heterogeneity, compared to a FE model!

3) Report the confidence intervals for I^2 (calculated using heterogi or metaan in Stata) as argued in http://www.ncbi.nlm.nih.gov/pubmed/17974687. A simple formula exists in the seminal 2002 Higgins paper that proposed I^2.

4) The authors argue that if heterogeneity is high they will not meta-analyse. Indeed, very high heterogeneity estimates according to some researchers mean studies should not be meta-analysed. However, I disagree with that assessment for numerous methodological reasons. First heterogeneity is not study size independent: smaller meta-analyses are more often homogeneous and larger studies are heterogeneous (for both methodological and practical reasons). So according to this mantra we will be filtering out the most valuable analyses, the ones that involve many studies, over the much smaller ones. In addition, random effects models can model that heterogeneity and account for it. Finally, large heterogeneity is the norm and it's great if it has been picked up and can be incorporated in the model. It is much more problematic when the underlying heterogeneity is not picked up and studies are "safely" combined under a homogeneity assumption. In other words, small meta-analyses of “homogeneous” studies are much more problematic than large “heterogeneous” ones as evidenced in this http://www.ncbi.nlm.nih.gov/pubmed/23922860. In smaller meta-analyses existing heterogeneity is just not picked up well enough. The uncertainty of the estimate becomes obvious if the CIs for I^2 are reported, as argued in http://www.ncbi.nlm.nih.gov/pubmed/17974687.

Minor

1) in search strategy, clarify that you will search "from inception" until the end date.

2) Some minor language corrections are needed.

3) Year may be worth considering in bias assessment, especially if you don't have enough studies for a formal test: http://www.ncbi.nlm.nih.gov/pubmed/25988604. With newer studies we would be more confident.

4) Clarify the weighting for the model for analysing dichotomous outcomes. Inverse variance (IV for random effects) or Mantel Haenszel (MH)? Note that MH is traditionally a fixed effect approach and the random effects version in RevMan is an IV-MH hybrid method. Anyway, clarification on the weighting is needed.

5) How will the random-effect model be implemented, i.e. how will heterogeneity be estimated? There are numerous ways to do so. Did they use the standard DerSimonian-Laird method? If so, please state so. Also there are better performing methods, for example please see https://www.ncbi.nlm.nih.gov/pubmed/28815652 (or http://www.ncbi.nlm.nih.gov/pubmed/23922860) and the metaan command in Stata where these are implemented (https://www.stata-journal.com/article.html?article=st0201).

6) Do you expect to have to use continuity corrections (adding 0.5 in cells to ensues zero event studies are included)? Please reflect on the recommendations in this paper: http://bmjopen.bmj.com/content/6/8/e010983.full and discuss or make changes in your analyses as appropriate. Maybe a sensitivity analysis is needed.

7) Cochran Q (i.e. chi-square) is notoriously underpowered to detect heterogeneity, especially for small meta-analyses http://www.ncbi.nlm.nih.gov/pubmed/9595615. I would not use.

Reviewers' comments:

Reviewer's Responses to Questions

**Comments to the Author**

1. Does the manuscript provide a valid rationale for the proposed study, with clearly identified and justified research questions?

Reviewer #1: Partly

Reviewer #2: Yes

Reviewer #3: Yes

Reviewer #4: Partly

2. Is the protocol technically sound and planned in a manner that will lead to a meaningful outcome and allow testing the stated hypotheses?

Reviewer #1: Partly

Reviewer #2: Yes

Reviewer #3: Yes

Reviewer #4: Partly

3. Is the methodology feasible and described in sufficient detail to allow the work to be replicable?

Reviewer #1: Yes

Reviewer #2: Yes

Reviewer #3: Yes

Reviewer #4: Yes

4. Have the authors described where all data underlying the findings will be made available when the study is complete?

Reviewer #1: Yes

Reviewer #2: Yes

Reviewer #3: Yes

Reviewer #4: Yes

5. Is the manuscript presented in an intelligible fashion and written in standard English?

Reviewer #1: Yes

Reviewer #2: No

Reviewer #3: Yes

Reviewer #4: Yes

6. Review Comments to the Author

You may also provide optional suggestions and comments to authors that they might find helpful in planning their study.

Reviewer #1: The paper was revised. Many sentences of the manuscript was rewritten. I have no significant comment.

Reviewer #2: I consider that the authors made a more detailed review after receiving the questions of the reviewers. The correction of English also allowed a better understanding of the manuscript.

Reviewer #3: The authors put more efforts in the manuscript. All of my concerns have been well addressed. It is suitable to publish in our journal now.

Reviewer #4: My previous comments are not included in the response, I wonder if the authors never received them. On the basis of my comments not being discussed, there is little for me to do but suggest rejection.

7. PLOS authors have the option to publish the peer review history of their article (what does this mean?). If published, this will include your full peer review and any attached files.

Reviewer #1: No

Reviewer #2: No

Reviewer #3: No

Reviewer #4: No

---

## [Author Response · Author response to Decision Letter 1]

9 Sep 2022

Dear Reviewers:

Thank you for your invaluable comments regarding our manuscript titled “Efficacy and safety of tibial transverse transport for diabetic foot: a protocol for systematic review and meta-analysis” (ID: PONE-D-21-40669). Those comments were all valuable and very helpful for revising and improving our paper, as well as serving as important guidelines for our researches. We have studied the provided comments carefully and made corrections for which we hope will be met with approval. We have uploaded the revised manuscript following the instructions provided in your letter. The revised sections are shown in red, and our responses to your comments are included below:

Response to Reviewers #1, #2, and #3:

Thank you very much for your time and consideration regarding the publication of our paper. On behalf of my co-authors, I would like to express our great appreciation to the reviewers.

Reviewer #4:

1.Comment: Define the outcomes that are not self explanatory. For example, I've never heard of "effective rate" before.

Response: We are grateful for your suggestion. In our previous pre-search, we found that the outcome index "effective rate" appeared in several literatures [1-3]. In order to more completely evaluate the clinical effect of tibial transverse transport for diabetic feet, we included "effective rate" as the outcome index. Several other indicators are similar. We also agree with your suggestion, therefore, we will improve the phrasing of relevant outcome indexes in the subsequent formal study, and look forward to your further suggestions.

[1] Du SJ, Li YN, Lai JY. Clinical efficacy of modified tibial transverse transport in the treatment of diabetic foot. Chinese Journal of Modern Drug Application. 2021;15(5):29-31. DOI:10.14164/j.cnki.cn11-5581/r.2021.05.009

[2] Jin AL. Clinical analysis of tibial transverse bone transport in the treatment of 

diabetic foot and lower extremity vascular occlusion. clinical research. Doctor. 2020;5(15):26-28.https://iffhc975c3444c2dc4daas5ubofkbvn0v96xw9fffi.res.gxlib.org.cn/periodical/ChlQZXJpb2RpY2FsQ0hJTmV3UzIwMjIwODI0Eg1keXNoMjAyMDE1MDEwGghtam11dzFpZA%3D%3D

[3]Ma JW, Zhao LL, Ma DC. Analysis of clinical efficacy of treatment of diabetic foot of lateral tibial bone removal microcirculation regeneration technology. Heilongjiang Science. 2018;9(9):6-7. DOI:10.3969/j.issn.1674-8646.2018.09.003.

2.Comment: Avoid fixed effect models since they under-perform in the presence of ANY heterogeneity. Random-effects (RE) models are more conservative and provide better estimates with wider confidence intervals: http://www.ncbi.nlm.nih.gov/pubmed/11252006 and http://www.ncbi.nlm.nih.gov/pubmed/21148194. What does the arbitrary 50% cut-off point add? Why is 49% fine and 51% an issue? My point is that a RE model will work better, in the presence of 5% heterogeneity, compared to a FE model!

Response: Thank you for underlining this deficiency. This section was revised according to your advice. With numerous underlying variables in medical studies, establishing homogeneity is a rare commodity and some degree of variability between studies should be anticipated [1]. A random effects model, however, allows the true effect to vary across studies. Random effects model builds in an estimate of the between-study variation in effect size and typically provides wider overall confidence intervals [2]. Brockwell SE et al. thought that the random effects model should be adopted irrespective of the I2 heterogeneity statistic outcome [3]. Therefore, we are very happy to accept your suggestion and use the random effects model for statistical analysis (Line 217-221, page 11-12).

[1] Thompson SG, Pocock SJ. Can meta-analyses be trusted? Lancet 1991; 338(8775): 1127–1130. 

[2] Kontopantelis E, Reeves D. Performance of statistical methods for meta-analysis when true study effects are non-normally distributed: A simulation study. Stat Methods Med Res. 2012;21(4):409-426. 

[3] Brockwell SE, Gordon IR. A comparison of statistical methods for meta-analysis. Stat Med. 2001;20(6):825-840. 

3.Comment: Report the confidence intervals for I^2 (calculated using heterogi or metaan in Stata) as argued in http://www.ncbi.nlm.nih.gov/pubmed/17974687. A simple formula exists in the seminal 2002 Higgins paper that proposed I^2.

Response: Thank you for your comment. We believe that all statistical tests for heterogeneity are weak, including I2. The clinical implications of this are considerable and must be examined on a case-by-case basis. Putting too much trust in homogeneity of effects may create a false sense of reassurance that one size fits all.  Given that I2 is not precise, 95% confidence intervals should always be given [1]. Therefore, we have set 95% confidence intervals for I2 (Line 222-223, page 12).

[1] Ioannidis JP, Patsopoulos NA, Evangelou E. Uncertainty in heterogeneity estimates in meta-analyses. BMJ. 2007;335(7626):914-6. doi: 10.1136/bmj.39343.408449.80. PMID: 17974687; PMCID: PMC2048840.

4.Comment: The authors argue that if heterogeneity is high they will not meta-analyse. Indeed, very high heterogeneity estimates according to some researchers mean studies should not be meta-analysed. However, I disagree with that assessment for numerous methodological reasons. First heterogeneity is not study size independent: smaller meta-analyses are more often homogeneous and larger studies are heterogeneous (for both methodological and practical reasons). So according to this mantra we will be filtering out the most valuable analyses, the ones that involve many studies, over the much smaller ones. In addition, random effects models can model that heterogeneity and account for it. Finally, large heterogeneity is the norm and it's great if it has been picked up and can be incorporated in the model. It is much more problematic when the underlying heterogeneity is not picked up and studies are "safely" combined under a homogeneity assumption. In other words, small meta-analyses of “homogeneous” studies are much more problematic than large “heterogeneous” ones as evidenced in this http://www.ncbi.nlm.nih.gov/pubmed/23922860. In smaller meta-analyses existing heterogeneity is just not picked up well enough. The uncertainty of the estimate becomes obvious if the CIs for I^2 are reported, as argued in http://www.ncbi.nlm.nih.gov/pubmed/17974687.

Response: Our deepest gratitude goes to you for your careful work and thoughtful suggestions that substantially helped improve our paper. Until we have received your advice, we believed that the very high heterogeneity means that studies should not be meta-analyzed. However, after reviewing the relevant data, we found that our previous view was indeed wrong, and some valuable research may be filtered out. With numerous underlying variables in medical studies, establishing homogeneity is a rare commodity and some degree of variability between studies should be anticipated [1]. When the number of studies required for meta-analysis is small, we often observe or assume that the variance estimate between studies is zero, and in fact this homogeneity assumption may be erroneous. Kontopantelis E et al. found levels of unobserved heterogeneity in the Cochrane data which appeared to be high, especially for small meta-analyses [2]. Whenever heterogeneity is detected, we do not ignore it, and we try to find its source (Line 230-233, page 12). 

[1] Thompson SG, Pocock SJ. Can meta-analyses be trusted? Lancet. 1991; 338(8775): 1127–1130. 

[2] Kontopantelis E, Springate DA, Reeves D. A Re-Analysis of the Cochrane Library Data: The Dangers of Unobserved Heterogeneity in MetaAnalyses. PLoS ONE. 2013;8(7): e69930. doi:10.1371/journal.pone.0069930

5.Comment:  In search strategy, clarify that you will search "from inception" until the end date.

Response: Thank you for your advice. We have added the full search date to the manuscript according to your advice (Line 30-31, page 2). 

6.Comment: Some minor language corrections are needed.

Response: Thank you for your careful review. We are very sorry for the mistakes in this manuscript and inconvenience they caused in your reading. The manuscript has been thoroughly revised and rewritten by Editage, a professional English language editing company, so we hope it can now meet the journal’s standard.

7.Comment: Year may be worth considering in bias assessment, especially if you don't have enough studies for a formal test: http://www.ncbi.nlm.nih.gov/pubmed/25988604. With newer studies we would be more confident.

Response: Thank you for the suggestion. The issue of publication bias in meta-analyses is often completely ignored. We have carefully read the literature you recommended. Kicinski Michal et al. found that publication bias is smaller in meta-analyses of more recent studies, indicating their better reliability, and supporting the effectiveness of the measures for reducing publication bias in clinical trials. In general, publication bias is larger in meta-analyses of older studies, indicating their lower reliability [1]. Publication year has an important impact on publication bias, and we hope to include more new studies in subsequent studies in order to reduce it.

[1]Kicinski Michal, Springate David A, Kontopantelis Evangelos. Publication bias in meta-analyses from the Cochrane Database of Systematic Reviews [J]. Stat Med, 2015;34:2781-2793.

8.Comment: Clarify the weighting for the model for analysing dichotomous outcomes. Inverse variance (IV for random effects) or Mantel Haenszel (MH)? Note that MH is traditionally a fixed effect approach and the random effects version in RevMan is an IV-MH hybrid method. Anyway, clarification on the weighting is needed.

Response: We deeply appreciate your suggestion. Accordingly, we have added a more detailed interpretation regarding the weighting of the model for dichotomous outcome analysis. We choose inverse variance to study the dichotomous outcomes. The study confidence interval and the total confidence interval are both 95%. We have added this information to the corresponding position in the manuscript (Line 210-212, page 11).

9.Comment: How will the random-effect model be implemented, i.e. how will heterogeneity be estimated? There are numerous ways to do so. Did they use the standard DerSimonian and Laird method? If so, please state so. Also there are better performing methods, for example please see https://www.ncbi.nlm.nih.gov/pubmed/28815652 (or http://www.ncbi.nlm.nih.gov/pubmed/23922860) and the metaan command in Stata where these are implemented (https://www.stata-journal.com/article.html?article=st0201).

Response: We agree with the comment and have expanded the sentence in the revised manuscript as follows: 

“It will be implemented using the standard DerSimonian-Laird (D-L) method (Line 221-223, page 12).

10.Comment: Do you expect to have to use continuity corrections (adding 0.5 in cells to ensues zero event studies are included)? Please reflect on the recommendations in this paper: http://bmjopen.bmj.com/content/6/8/e010983.full and discuss or make changes in your analyses as appropriate. Maybe a sensitivity analysis is needed.

Response: Thank you for your suggestion. Friedrich recommended including zero event studies in all meta-analyses for the benefit of providing conservative point estimates and increasing the study integrity [1]. Cheng et al. found that including zero event studies provided more accurate overall pooled estimates than excluding them when there was no true treatment effect [2]. According to your comment, we have added the suggested content to the manuscript as follows: 

“...if conditions permit, a continuity correction factor will be added to each of the four cells of the 2×2 table for zero event studies, i.e., to the event and non-event in the treatment arms, and event and non-event in the control arms. The constant continuity factor will be 0.5 (Line 233-237, page 12).”

[1] Friedrich JO, Adhikari NKJ, Beyene J. Inclusion of zero total event trials in meta-analyses maintains analytic consistency and incorporates all available data. BMC Med Res Methodol 2007;7:5.

[2] Cheng J, Pullenayegum E, Marshall JK, Iorio A, Thabane L. Impact of including or excluding both-armed zero-event studies on using standard meta-analysis methods for rare event outcome: a simulation study. BMJ Open. 2016 Aug 16;6(8):e010983. doi: 10.1136/bmjopen-2015-010983. PMID: 27531725.

11.Comment: Cochran Q (i.e. chi-square) is notoriously underpowered to detect heterogeneity, especially for small meta-analyses http://www.ncbi.nlm.nih.gov/pubmed/9595615. I would not use.

Response: We are very grateful to you for pointing out this problem. Paul and Donner found the power of the Cochran's Q test to be lower than those of other homogeneity tests for odds ratios in k 2x2 tables [1]. Hardy and Thompson found that the power of the Cochran’s Q test for heterogeneity can be low, especially when the total available information is low, i.e., in the case of sparse data [2]. After consulting the relevant materials, we found that Cochran's Q test usually has low power for detecting heterogeneity. Therefore, we will not use it for testing heterogeneity in future studies.

[1] Paul SR, Donner A. Small sample performance of tests of homogeneity of odds ratios in k 2x2 tables. Stat Med. 1992 Jan 30;11(2):159-65. doi: 10.1002/sim.4780110203. PMID: 1579755.

[2] Hardy RJ, Thompson SG. Detecting and describing heterogeneity in meta-analysis. Stat Med. 1998 Apr 30;17(8):841-56. doi: 10.1002/(sici)1097-0258(19980430)17:8<841::aid-sim781>3.0.co;2-d. PMID: 9595615.

Thank you for your careful review. We really appreciate your efforts in reviewing our manuscript during these unprecedented and challenging times. We wish good health to you, your family, and community. Your careful review has helped make our study clearer and more comprehensive. Some changes that we have made will not influence the content and framework of the paper, and we have thus not listed them here, but marked them in red in the revised text. We trust that the revised manuscript will be suitable for publication in your journal.

Sincerely,

Guanwen Sun

Email: sunguanwen@sina.com

---

## [Decision Letter · Decision Letter 2]

25 Oct 2022

Efficacy and safety of tibial cortex transverse transport for diabetic foot: a protocol for systematic review and meta-analysis

PONE-D-21-40669R2

Dear Dr. Sun,

We’re pleased to inform you that your manuscript has been judged scientifically suitable for publication and will be formally accepted for publication once it meets all outstanding technical requirements.

Kind regards,

Liang-Tseng Kuo, M.D.

Academic Editor

PLOS ONE

Additional Editor Comments (optional):

Reviewers' comments:

Reviewer's Responses to Questions

**Comments to the Author**

1. Does the manuscript provide a valid rationale for the proposed study, with clearly identified and justified research questions?

Reviewer #1: Yes

Reviewer #2: Yes

Reviewer #3: Yes

Reviewer #4: Yes

2. Is the protocol technically sound and planned in a manner that will lead to a meaningful outcome and allow testing the stated hypotheses?

Reviewer #1: Yes

Reviewer #2: Yes

Reviewer #3: Yes

Reviewer #4: Yes

3. Is the methodology feasible and described in sufficient detail to allow the work to be replicable?

Reviewer #1: Yes

Reviewer #2: Yes

Reviewer #3: Yes

Reviewer #4: Yes

4. Have the authors described where all data underlying the findings will be made available when the study is complete?

Reviewer #1: Yes

Reviewer #2: Yes

Reviewer #3: Yes

Reviewer #4: Yes

5. Is the manuscript presented in an intelligible fashion and written in standard English?

Reviewer #1: Yes

Reviewer #2: Yes

Reviewer #3: Yes

Reviewer #4: Yes

6. Review Comments to the Author

You may also provide optional suggestions and comments to authors that they might find helpful in planning their study.

Reviewer #1: The reviewer agree that the paper is accepted. The reviewer does not have any further comments for the author response.

Reviewer #2: Dear author, I found that all the questions and suggestions asked by the reviewers were important for the technical improvement of the article. I believe that the results of this analysis will be much more consistent.

Reviewer #3: The authors have already addressed my comments. I have no further comments regarding this manuscript and do believe our journal should get it in. Thank you!

Reviewer #4: I am quite satisfied with the authors' responses and resulting changes to the manuscript and I have nothing further to add.

7. PLOS authors have the option to publish the peer review history of their article (what does this mean?). If published, this will include your full peer review and any attached files.

Reviewer #1: No

Reviewer #2: No

Reviewer #3: No

Reviewer #4: No

---

## [Editor Report · Acceptance letter]

27 Oct 2022

PONE-D-21-40669R2 

Efficacy and safety of tibial cortex transverse transport for diabetic foot: a protocol for systematic review and meta-analysis 

Dear Dr. Sun:

I'm pleased to inform you that your manuscript has been deemed suitable for publication in PLOS ONE. Congratulations! Your manuscript is now with our production department. 

Kind regards, 

on behalf of

Dr. Liang-Tseng Kuo 

Academic Editor

PLOS ONE